# A Narrative Review on the Approach to Antimicrobial Use in Ventilated Patients with Multidrug Resistant Organisms in Respiratory Samples—To Treat or Not to Treat? That Is the Question

**DOI:** 10.3390/antibiotics11040452

**Published:** 2022-03-27

**Authors:** Lowell Ling, Wai-Tat Wong, Jeffrey Lipman, Gavin Matthew Joynt

**Affiliations:** 1Department of Anaesthesia and Intensive Care, The Chinese University of Hong Kong, Hong Kong, China; lowell.ling@cuhk.edu.hk (L.L.); wtwong@cuhk.edu.hk (W.-T.W.); 2Royal Brisbane and Women’s Hospital and Jamieson Trauma Institute, The University of Queensland, Brisbane, QLD 4029, Australia; j.lipman@uq.edu.au; 3Nimes University Hospital, University of Montpelier, 30900 Nimes, France

**Keywords:** ventilator associated pneumonia, ventilator associated tracheobronchitis, ICU, treatment, decision, multidrug resistant

## Abstract

Multidrug resistant organisms (MDRO) are commonly isolated in respiratory specimens taken from mechanically ventilated patients. The purpose of this narrative review is to discuss the approach to antimicrobial prescription in ventilated patients who have grown a new MDRO isolate in their respiratory specimen. A MEDLINE and PubMed literature search using keywords “multidrug resistant organisms”, “ventilator-associated pneumonia” and “decision making”, “treatment” or “strategy” was used to identify 329 references as background for this review. Lack of universally accepted diagnostic criteria for ventilator-associated pneumonia, or ventilator-associated tracheobronchitis complicates treatment decisions. Consideration of the clinical context including signs of respiratory infection or deterioration in respiratory or other organ function is essential. The higher the quality of respiratory specimens or the presence of bacteremia would suggest the MDRO is a true pathogen, rather than colonization, and warrants antimicrobial therapy. A patient with higher severity of illness has lower safety margins and may require initiation of antimicrobial therapy until an alternative diagnosis is established. A structured approach to the decision to treat with antimicrobial therapy is proposed.

## 1. Introduction

Ventilator-associated pneumonia (VAP) is one of the most common nosocomial infections in the intensive care unit (ICU) [1]. Ventilator-associated tracheobronchitis (VAT) is considered a distinct precursor to VAP [2,3]. Both cause significant morbidity to mechanically ventilated patients [2,4,5,6]. VAP/VAT are associated with increased ventilator days, ICU and hospital length of stay [2,7]. The attributable mortality of VAP is around 3–17%, whilst risk of mortality from VAT remains controversial [8,9]. Furthermore, VAP incurs a huge cost to healthcare systems [7]. The incidence of VAP varies widely between 1.2 and 116 per 1000 ventilator days and is likely due to differences in case mix, diagnostic criteria, and the purpose of local surveillance programs [10,11].

Worldwide, the burden of multidrug resistant organisms (MDRO) is rising [12]. Antimicrobial resistance (AMR) caused 1.27 million deaths globally in 2019. The situation is no different in the ICU, which has high rates of MDRO infections. Common MDROs in the respiratory tract include multidrug resistant *Pseudomonas aeruginosa* and *Acinetobacter baumannii*, which require treatment with antimicrobial agents such as tigecycline or colistin [13]. Although the prevalence and type of MDRO-related VAP/VAT varies across different health settings [3,14,15,16], patients who have VAP due to MDRO generally have worse survival than patients with sensitive pathogens, especially when inappropriate antibiotics are used [17,18,19]. Furthermore, there are indications that the incidence of VAP caused by MDRO has increased during the current COVID-19 pandemic [20]. 

In the setting of a positive microbiological result, the problem of distinguishing VAP/VAT from colonization is difficult, and the lack of a universally accepted diagnostic criteria for VAP/VAT also complicates treatment decisions [6,21]. Nevertheless, the use of antimicrobial therapy for VAT is associated with reduction in progression to VAP, whilst early appropriate treatment in VAP is associated with improved survival [22,23]. Unfortunately, there are also a number of well-established harms associated with the use of antimicrobial agents in ICU [24]. Most importantly, the excessive or inappropriate use of broad-spectrum antibiotics, particularly in patients who do not have VAP, potentially promotes AMR [25]. The inappropriate use of antimicrobials negatively affects both the individual and other patients [24]. Antimicrobial use in critically ill patients has been shown to be associated with multiple drug toxicities, as well as AMR [24,26,27]. Specifically, studies have shown that the injudicious use of broad-spectrum antimicrobials for ICU-acquired infections may be associated with higher prevalence of multidrug resistant organisms and increased mortality [28,29]. While the clinical question in these studies examined the difference in outcomes between initiation of broad-spectrum antibiotics versus delayed but targeted antimicrobials, it seems plausible that withholding broad-spectrum antimicrobials in patients with likely MDRO colonization, rather than starting inappropriate broad-spectrum antimicrobial therapy, would have similar benefits.

Recent reviews have provided comprehensive summaries of epidemiological data and the recommended empirical treatment of patients with suspected or confirmed MDRO VAP/VAT [21,30]. However, we believe none explicitly detail and integrate the different elements of the clinical decision process required to determine whether antimicrobials should be initiated in ventilated patients with known MDRO respiratory isolates. Thus, the purpose of this narrative review is to discuss the relevant literature supporting the specific set of conditions that must be considered when deciding to initiate antimicrobial therapy in ventilated patients who have grown a new MDRO isolate in their respiratory specimens. As a result of this review, a comprehensive clinical framework is proposed as a guide to facilitate decisions on antimicrobial use in this setting. 

## 2. Literature Search

A literature search was performed in MEDLINE and PubMed for published studies since 1910 on 14 March 2022 using the strategy described in Appendix A. The total number of articles identified was 418, of which 89 were duplicates. All authors independently reviewed the abstract of 329 articles for inclusion and utilized the full text of 2 articles considered directly relevant to this narrative review. Following the review of references of identified articles, 80 additional articles were included. These additional articles were added after consensus with all authors; in the case of disagreement, the corresponding author made the final decision.

## 3. Definition of VAP/VAT

Establishing whether a patient has VAP/VAT is one of the essential steps to establish the relevance of respiratory MDRO isolates. Patients who are colonized with MDRO do not need treatment and prescription of antimicrobials targeting the MDRO may worsen AMR in this setting [24]. The classic clinical definition of VAP includes infiltrates on a chest radiograph with two of either fever, leucocytosis, or purulent sputum [31]. Unfortunately, this criterion only has a sensitivity of 69% and specificity of 75% when lung histology and culture is used as gold standard. However, it should be noted that there is considerable disagreement even amongst histopathologists on what constitutes pneumonia on histology [32].

Most individual signs lack effective diagnostic accuracy. For example, an individual sign such as fever only has 66.4% sensitivity and 53.9 specificity to diagnose VAP [33]. Similarly, leucocytosis alone is non-specific and only has 64.2% sensitivity and 59.2 specificity in identifying VAP. Furthermore, while infiltrates on plain chest radiographs are often non-specific for pneumonia, even the absence of visible infiltrates lacks the sensitivity to effectively rule out VAP [34,35]. Therefore, the Clinical Pulmonary Infection Score (CPIS) was proposed more than 30 years ago to improve the diagnosis of VAP [36]. Whilst individual components of fever, white cell count, tracheal secretions, PaO_2_/FiO_2_ ratio, microbiological sampling, and chest radiography each has poor sensitivity and specificity for VAP, the hope was that a composite score would enhance diagnostic performance. Yet, when tested, the CPIS only had 65% sensitivity and 64% specificity to diagnose VAP [37]. 

An alternative diagnostic approach is to use the Possible Ventilator-Associated Pneumonia (PVAP) surveillance criteria proposed by the Centers for Disease Control and Prevention (CDC) in the United States [38]. The diagnostic algorithm starts with identification of a ventilator associated event defined as a 2-day escalation of FiO_2_ or PEEP after a stable or decreasing trend for 2 days. If the patient has a temperature >38 or <36 °C, or white cell count ≥ 12,000 or ≤4000 cells/mm^3^ and a new antimicrobial is started for at least 4 days, then the patient has an infection-related ventilator-associated complication. Subsequently if the patient has a pathogen or histological evidence of pneumonia confirmed within 2 days of worsening oxygenation, then the patient is classified as having PVAP. However, because antimicrobial prescription is itself in the CDC definition, it is not possible to use the definition to help guide decisions on antimicrobial initiation. Furthermore, the CDC definition has low sensitivity for VAP which limits its utility as a clinical screening tool [39,40]. 

Because of the high prevalence of VAP in intensive care units, plain chest radiographs are often performed to screen for pneumonia in ventilated patients. However, the diagnostic performance of plain radiographs for VAP is poor, with a sensitivity of 24–60%, and specificity of 29–91% [41,42]. This means that plain chest radiographs are insufficiently sensitive to detect VAP. Indeed, computed tomography (CT) of the lung has better sensitivity and specificity and can improve the diagnostic yield for VAP when compared to plain radiographs [43]. However, CT scanners generally lack portability, have a substantially higher radiation exposure dose, and are inconvenient as a generic screening tool. When used selectively, CT scans may be useful in selected cases where plain chest radiographs are clear, and the significance of MDRO growth in the respiratory tract is doubtful. In this setting, CT thorax should be performed to improve diagnostic accuracy and improve the precision of the decision to treat the MDRO. The use of other lung imaging technologies, such as point of care lung ultrasound are promising investigative modalities, but current evidence on its diagnostic performance on VAP is sparse [44,45]. 

Similar to VAP, a universally accepted definition of VAT is lacking [21]. The commonly accepted criteria to diagnose VAT are similar to those of VAP, but with the absence of radiological evidence of pneumonia [46]. Since VAT is also associated with increased ventilator days and lengthened ICU stay, the diagnostic boundary and clinical significance of differentiating between VAT and VAP is somewhat unclear [2,3]. However, as discussed above, radiological absence of pneumonia cannot rule out VAP, and it is possible that previous studies were not sufficiently rigorous to differentiate the two entities. Whether treatment should be initiated for VAT is contentious. A detailed discussion on the merits and methods of treating VAT is beyond the scope of this review and was addressed in a recent comprehensive review [21]. 

The absence of definitive clinical diagnostic criteria for VAT, and as for VAP, means that treatment decisions must be made on an individual patient basis. In this review, we describe the factors that clinicians should consider in a tailored and balanced approach to determine the need for antimicrobial therapy in patients who have the presence of MDRO reported in respiratory specimens.

## 4. Characteristics of Respiratory Specimens

The type, quality, and timing of respiratory specimens are key factors to consider when interpreting microbiological results. Most of our understanding of microbiological sampling of the lower respiratory tract comes from clinical studies of VAP rather than from VAT. The interpretation of results for the latter is poorly described, but often assumed to be the same for VAP [47]. In both cases, specimens with >10 squamous epithelial cells should be interpreted as being the consequence of contamination from upper airway flora [48]. In parallel, distal sampling by bronchoscopy is thought to be more representative of the presence of true pathogens in the lung. In addition, endotracheal tube biofilms develop shortly after intubation and may harbor common pathogenic organisms; thus, non-invasive sampling for diagnosis of VAP via the endotracheal tube may potentially result in false positive growth secondary to contamination [49]. Nevertheless, the available clinical evidence that compares the diagnostic accuracy of bronchoscopic specimens with endotracheal samples for diagnosis of VAP remains inconclusive [50,51]. 

Quantitative cultures are often considered to have better diagnostic accuracy for VAP when compared to qualitative or even semi-quantitative cultures [52]. This is in part because the endotracheal tube and respiratory tract are colonized quickly after intubation [53], and quantitative cultures may be more capable of distinguishing lung parenchymal growth from upper respiratory tract contamination. Comparative studies comparing bronchoscopic and tracheal samples against the gold standard of lung biopsy have established specific thresholds that are suggestive of pathogenic growth in pneumonia (≥10^4^ colony forming units/mL for bronchoscopic specimens, and ≥10^5^ colony forming units/mL for tracheal aspirates) [54]. Using these thresholds may reduce, but is unlikely to eliminate the risk of attributing culture results to pathogenic growth rather than less harmful colonization of the upper airway. However, clinical evidence supporting this practice is limited [55,56]. One randomized controlled trial (RCT) found that the use of quantitative cultures from BAL or brushing compared to non-quantitative tracheal aspirates was associated with less antibiotic use and even improved survival [57]. However, another RCT concluded that patients with suspected VAP had equivalent outcomes and similar antibiotic use, regardless of whether quantitative BAL or non-quantitative tracheal aspirates were performed to guide antimicrobial therapy [58]. 

An important caveat is that investigations alone do not improve outcomes unless they are associated with appropriate therapeutic interventions. Observational data suggest that discontinuation of antibiotics based on quantitative culture thresholds is safe, reduces antibiotics duration, and reduces MDRO infections [59,60,61]. In patients with MDRO isolated from respiratory specimens, we recommend starting or ending targeted treatment based on semiquantitative or quantitative cultures only, and not relying on qualitative cultures alone, since the risk of subsequent increased or pan-resistance is ominous. The MDRO result should be interpreted in context of the patient’s clinical picture. Growth of MDRO despite clinical improvement may be cautiously discounted as pathogenic. Conversely, clinical deterioration with an MDRO respiratory specimen after an initial improvement in the original pathology, is more likely to suggest new infection. 

All diagnostic tests should be interpreted on the basis of pre-test probability. The higher the pre-test probability, the greater the likelihood that a positive test indicates the true presence of the condition. The pre-test probability of VAP/VAT in the presence of a positive MDRO culture is very different when a sample was taken because of clinically suspected VAP/VAT when compared with the result of screening cultures taken for low grade fever, or for pan-surveillance cultures to establish the nature of local microbiological flora. In the former group, if multiple positive cultures are returned, the clinician will need to decide which pathogen is likely to explain the clinical picture. For example, if the patient also has bacteremia with the MDRO then treatment should be initiated, although only a minority of patients with VAP have concurrent bacteremia [62]. Nevertheless, antimicrobial therapy is likely necessary. On the other hand, in the latter case positive surveillance cultures in the absence of clinical findings of VAP should likely be interpreted as colonization because of the very low pre-test probability of current VAP. Furthermore, clinical assessments of current severity and the trend of severity of illness forms a key aspect of a patient’s clinical picture, and assignment of likely causality of the MDRO pathogens grown. 

## 5. Severity of Illness

Changes in pulmonary and extra-pulmonary organ dysfunction may provide clues to determine whether the MDRO in the respiratory tract is responsible for VAP or is simply a colonizing growth. Again, the major determining factor is the pre-test probability of VAP, making the assumption that VAP causes a deterioration in pulmonary function. Key parameters to assess the severity and trajectory of pulmonary function include: PaO_2_/FiO_2_ (PF ratio), peak end-expiratory pressure (PEEP), driving pressure, and dsynchrony. A worsening PF ratio and PEEP that persists for more than 2 days, and occurring within 2 days of the MDRO growth, suggest it is responsible for the deterioration in gas exchange and ventilatory support required [38]. This strategy may minimize the chance of falsely attributing transient worsening of respiratory failure from atelectasis or sputum retention to VAP. Similarly, higher driving pressures, lower compliance, and need for initiation or escalation of sedation and paralysis for ventilator synchrony may suggest new VAP. 

Whilst a deteriorating trend in pulmonary parameters is the expected clinical course when new VAP occurs, the diagnosis of VAP and antimicrobial treatment may also be warranted in patients with persistent respiratory failure, particularly when alternative diagnoses have been ruled out. The decision to treat the MDRO also depends on the physiological reserve of the patient. A patient who requires 80% FiO_2_ while receiving controlled ventilation and high levels of positive end expiratory pressure despite sedation and paralysis, has little reserve and room for diagnostic error. On the basis of likely risks and benefit, there may be no choice but to treat the MDRO specimen in such a patient as this may be the last opportunity to improve survival. On the contrary, a “wait and see” approach may be warranted in less ill patients to allow more time for observation and investigation, and in this way minimize overly liberal use of multiple and/or broad-spectrum antimicrobials. 

Changes in extra-pulmonary organ function may also give clues as to whether the MDRO should be treated as being causative of VAP. Worsening shock with the need for progressively increased vasopressor dose, or the need for frequent fluid bolus administration, suggest a deterioration in hemodynamic function and the likelihood of systemic sepsis. Progressive acute kidney injury, hyperbilirubinemia, or thrombocytopenia also signal new onset organ dysfunction from sepsis [63]. In this context, even if there are few, or subtle clinical, radiological, or microbiological clues that suggest VAP; then treatment of the MDRO would be prudent to avoid catastrophic and irreversible deterioration. Nevertheless, attributing organ dysfunction to infection may be difficult as there is considerable interobserver variability in recognition of sepsis amongst intensivists [64]. In fact, 13% of ICU patients initially treated for sepsis do not have underlying infection [65].

An equally challenging diagnostic dilemma may occur when deterioration of respiratory function is suspected to be from secondary acute respiratory distress syndrome from an extra-pulmonary infectious or non-infectious cause. In this setting, treatment must be directed to the most likely underlying cause, although coverage of the MDRO may be required concurrently. The use of quantitative bronchoscopic cultures to confirm the diagnostic significance of MDRO in this setting should be considered. 

Lastly, as a final consideration, VAP itself is not always associated with sepsis or overt organ dysfunction. It has been reported that only 30% of VAP patients have septic shock [66]. Thus, it remains necessary to treat the MDRO if there are convincing clinical signs of localized pulmonary infection as described above, even without clear evidence of systemic sepsis or septic shock. 

## 6. Infection and Inflammatory Markers

Standard clinical observations may reveal persistent leucocytosis or leucopenia, and high fever that are suggestive of infection. Tachycardia and high minute ventilation signify raised metabolic rate, often caused by systemic inflammation or infection. However, as discussed previously, none of these findings are sufficiently sensitive nor specific, either singly or in combination to secure a diagnosis with a high degree of certainty. 

A definitive biomarker able to confirm the presence of bacterial infection or sepsis with a high diagnostic performance remains elusive, despite decades of research. To date, few biomarkers have demonstrated clinical utility, and among these, the use of procalcitonin is best supported by current evidence, particularly for determining the presence of bacterial infection [67]. Studies have shown that protocolized use of procalcitonin in ICU patients with suspected infection is associated with a reduction in antibiotic duration, without causing either adverse microbiological or patient-centered outcomes [68,69,70]. Furthermore, there is increasing evidence demonstrating that the use of procalcitonin to inform the cessation of antimicrobial therapy in septic ICU patients may reduce the risk of subsequent MDRO infections, and possibly reduce mortality [71]. There is also specific data to show that procalcitonin-guided antimicrobial use reduces antibiotic exposure in patients with VAP, without any associated adverse effects [72]. 

Procalcitonin may be potentially useful in patients where the diagnosis of VAP is uncertain in the presence of MDRO growth, but antibiotic therapy is considered desirable on a balance of the benefit–risk assessment. In this setting, appropriate broad-spectrum antibiotics should be given first, with a plan to stop if procalcitonin levels are persistently low, or an alternative diagnosis established. Nevertheless, procalcitonin levels should always be interpreted in conjunction with other clinical factors, as diagnostic yield of PCT is limited if tested too early in the course of a new infection, or if the infection remains localized. False positives may be caused by non-infective inflammatory conditions; however, in general, procalcitonin is more specific for bacterial infections than other inflammatory markers. Procalcitonin may have an enhanced role in patients with comorbidities such as malignancy and immunosuppression, which may mask typical clinical signs of infection, although the current evidence supporting use in this setting remains scarce, and to some degree conflicting [73,74,75]. 

## 7. Comorbidities

Patients’ comorbidities may affect the interpretation of microbiological results and complicate the diagnosis of VAP. Chronic lung disease such as chronic obstructive pulmonary disease or bronchiectasis may predispose patients to respiratory tract colonization with MDRO because of the altered lung microbiological defense mechanisms, and the frequent exposure of such patients to antibiotics [76,77,78]. Patients residing in nursing homes may also be chronically colonized with MDRO [79]. In these situations, MDRO growth in respiratory samples in the absence of clinical evidence of deteriorating lung function or sepsis, makes infection less likely.

In contrast, patients who are immunosuppressed, either from hematological disease or chemotherapy, may not exhibit the classical clinical signs of pneumonia [80]. Patients may be afebrile without purulent sputum, or productive cough, and may not exhibit signs of pneumonia on examination. Furthermore, classic radiological signs of consolidation may be absent. Neutropenia may reduce the purulence of tracheal aspirates in patients who have VAP, although the evidence that this is consistently the case is lacking. Procalcitonin concentrations correlate with the presence of bacterial infection and the severity of infection in this population, but as baseline concentrations are variable, optimal concentrations have not been defined, and its use to guide antimicrobial use has not yet been adequately determined. On the contrary, patients with inflammatory airway diseases such as asthma may have chronically inflamed airways with raised neutrophil counts in the absence of pneumonia [81]. 

## 8. Alternative Diagnoses

Patients with suspected VAP may turn out to have alternative diagnoses [82,83]. Treatment thresholds are often higher for patients with MDRO growth in respiratory specimens than for patients with sensitive pathogens, because the need to use broad-spectrum antibiotics raises the risks of emergence of pan-resistant organisms, drug toxicity, drug cost, and future antibiotic option availability. Therefore, diagnostic precision should be maximized, and alternative diagnoses actively ruled out, to increase the likelihood of appropriate antimicrobial use for VAP. Thorough clinical examination and targeted investigations with echocardiography, CT and ultrasound of the thorax, abdomen, and pelvis should be considered to seek alternative diagnoses, particularly if the infective significance of the MDRO growth is uncertain within the clinical context. 

## 9. A Practical Approach to Management

We recommend an individualized and tailored approach to the interpretation and management of MDRO growth in respiratory specimens. The decision to treat or not to treat should be informed by the pre-test probability of VAP (clinical picture and diagnostic confidence), the nature of the specimen, and the risk of rapid deterioration if antimicrobials are withheld. The decision should be one supported by reasonable certainty that VAP exists, as the potential benefits of antimicrobial treatment must be balanced against the costs of antimicrobial therapy that not only includes monetary considerations, but also the many potential antimicrobial drug side-effects, and the substantial risk of worsening AMR, both for the individual patient and the environment (Figure 1). At one end of the spectrum, patients who clearly manifest signs of infection with swinging fever, leucocytosis, and radiological signs of VAP should probably be started or continued with targeted antimicrobial therapy for the MDRO. This is even more important for patients with minimal physiological reserve and that have high risk of death, as delay in VAP treatment may be fatal [22,84,85]. At the other end of the spectrum, patients who have grown an MDRO in a respiratory sample taken for surveillance despite clinical improvement and do not exhibit signs of infection can likely be safely observed expectantly, as the MDRO is likely a colonizing organism. If such a decision is made, continued clinical vigilance is required, as the patient may progress to develop VAT or VAP. If VAT is diagnosed, early and appropriate broad-spectrum antimicrobial therapy may be justified to reduce progression to VAP [23,86]. 

Patients who fall between these two extremes present the greatest treatment dilemma. The accompanying algorithm is designed to provide a broad framework covering the key decision-making factors that contribute to final decision making regarding whether to treat or not to treat (Figure 2). Where there is continued diagnostic uncertainty, often associated with a restricted choice of antimicrobials in the setting of multi, or pan-resistant organisms, and moderate severity of illness, we recommend the following pragmatic approach. We recommend starting treatment with appropriate broad-spectrum antimicrobial therapy, but with reassessment of the treatment plan at a pre-determined time (usually within 2–3 days). Such reassessment should include a review of procalcitonin concentration trends, a detailed clinical assessment, and other imaging and investigations as necessary to rule out other non-VAP/VAT pathology.

## 10. Limitations

There are several limitations to narrative reviews that include the subjective nature of the determination of which studies to include, the way the studies are interpreted, and the conclusions drawn. Nevertheless, our literature search showed that there is limited literature specifically on whether treatment should be given when MDROs are isolated from ventilated patients’ respiratory samples. Specifically, our extensive search did not reveal any high-level evidence to guide optimal decision making that encompasses and balances all the factors discussed in this review. While a systematic review of current literature is unlikely to directly answer the question of antibiotic initiation decision making, future work should focus on capturing empirical data that can quantify the magnitude of the positive and negative outcomes of antimicrobial treatment in ventilated patients with MDRO in respiratory samples, as summarized in Figure 1 and Figure 2. Although this was not a systematic review with rigorous data extraction by independent reviewers, our aim was to synthesize current best knowledge to provide a clinical framework for facilitating frontline physicians to make the best decisions on the need for initiation of antimicrobial use in ventilated patients with MDRO respiratory isolates.

## 11. Conclusions

MDRO are commonly isolated from respiratory specimens of mechanically ventilated patients. Treatment decisions for suspected VAP or VAT should be tailored to clinical picture, nature of the specimen, diagnostic confidence, balanced against the cost and risk of worsening AMR. We were unable to identify any clinically focused guidelines directly addressing all aspects of the complex decision-making process required to determine whether the presence of a MDRO specimen requires antimicrobial treatment initiation. We therefore suggest a multi-consideration approach, summarized in a simplified algorithm, to assist clinical decision making.

## Figures and Tables

**Figure 1 antibiotics-11-00452-f001:**
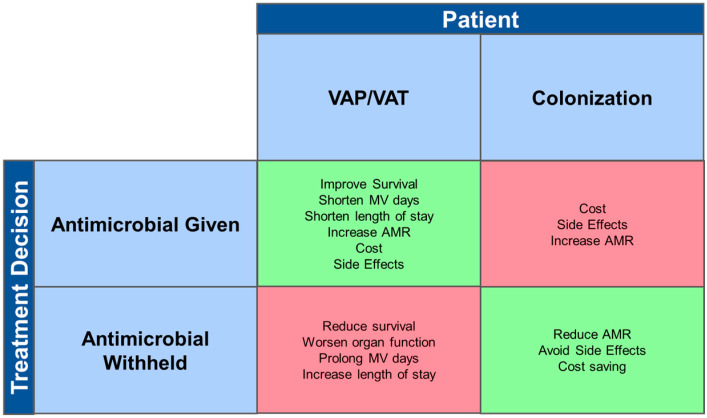
Risk and benefits of giving and withholding treatment for patients with VAP/VAT or respiratory colonization with MDRO. Red boxes represent consequences when decision to treat or not to treat is incorrect. Green boxes represent best case scenarios even when MDRO is isolated from respiratory samples of ventilated patients. AMR, antimicrobial resistance; MV, mechanical ventilation; VAP, ventilator-associated pneumonia; VAT, ventilator-associated tracheobronchitis.

**Figure 2 antibiotics-11-00452-f002:**
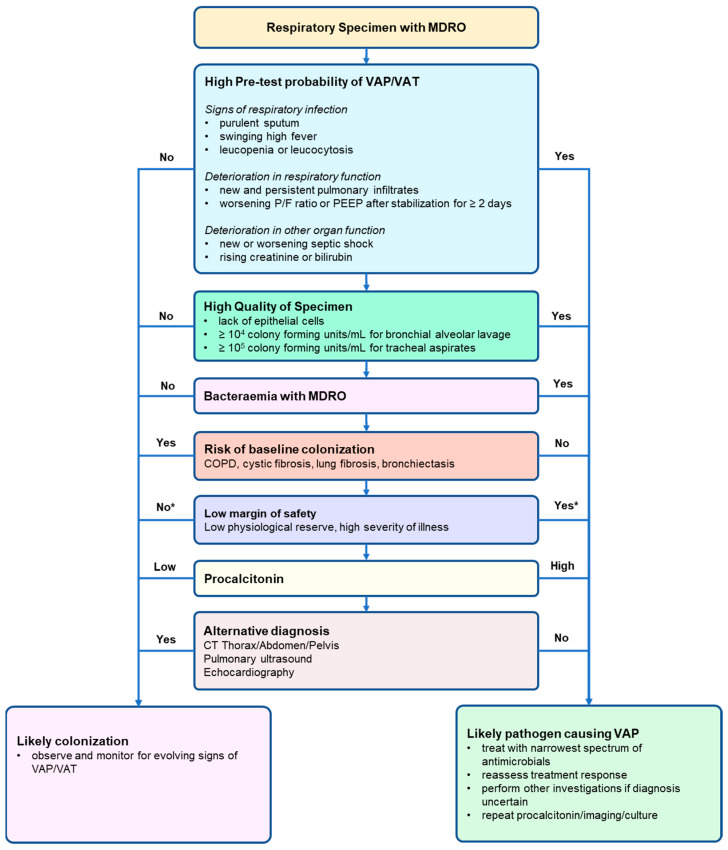
Outline of key factors to be considered when deciding to treat or not to treat when a ventilated patient has a positive MDRO respiratory specimen. The final decision is based on the final weight of evidence either in favor of likely VAP or not, after systematic consideration of these multiple factors. See text for further explanation. * Does not change the likelihood of VAP diagnosis but does reduce the safety margin of withholding targeted antimicrobial therapy. MDRO, multidrug resistant organism; PEEP, positive end-expiratory pressure; P/F, PaO_2_/FiO_2_; VAP, ventilator-associated pneumonia; VAT, ventilator-associated tracheobronchitis.

## Data Availability

Not applicable.

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
