# Peer review of "A Narrative Review on the Approach to Antimicrobial Use in Ventilated Patients with Multidrug Resistant Organisms in Respiratory Samples—To Treat or Not to Treat? That Is the Question"

_antibiotics, 2022, doi:10.3390/antibiotics11040452_

Round 1
Reviewer 1 Report
I deeply appreciate the opportunity to review the manuscript titled “To treat or not to treat? That is the question: ventilated patients with multidrug resistant organisms in respiratory specimens”. The authors present a narrative review that aims to “discuss the approach to antimicrobial prescription in ventilated patients who have grown a new MDRO isolates in their respiratory specimens”.
Overall, the manuscript is clear and easy to read. The authors address an interesting topic, especially considering the current COVID-19 pandemic. However, it is unclear to me why is this narrative review required, when the amount of primary evidence supports the conduction of a systematic review. Furthermore, the authors fail to address several other reviews that were published recently on this same topic. Below are my main concerns with the manuscript and suggestions for future revisions.
[Title, abstract and keywords]
- Although I do enjoy the spin on Shakespeare’s famous quote, current international guidelines recommend that authors clearly state the study design in the title. This increases potential article finding and citation.
- The abstract is lacking a more detailed approach to the method, please consider revising it (e.g., used mnemonic, databases/repositories searched, terms, filters, etc.).
- The abstract does not indicate how many articles were included in this review or the major topics addressed by the included studies. Please revise.
- I would argue that the keywords “critically ill”, “treatment” and “decision” are not required.
[Introduction]
- After reading the introduction section, it is still unclear to me why is this narrative review required – what is the novelty here?
- The authors fail to mention several recent reviews that tackle similar topics such as Chaïbi et al. (2022; doi: https://doi.org/10.3390/antibiotics11030359), Carrara et al. (2018; doi: 10.1016/j.ijantimicag.2017.12.013), Abu-Rub et al. (2021; doi: 10.3390/antibiotics10080935), or Pasero et al. (2021; doi: 10.3390/microorganisms9081773). The contributions of this review to existing evidence are lacking.
[Method]
- There is no indication of a method section, which is rather confusing for a narrative review. A convincing narrative review must be transparent about the sources of information on which the text is based. This is not the case here. Please revise.
- The authors fail to provide any statement concerning the database/repositories, search strategy, or filters used. Likewise, it is unclear if study selection, data extraction and report were conducted by a single member or more.
- For more information on what is expected in a Narrative Review, please see SANRA - the Scale for the Assessment of Narrative Review Articles, by Baethge et al. (2019, doi: 10.1186/s41073-019-0064-8).
[Results, discussion, and conclusion]
- What is the purpose of the second section (Definition of VAP)? I believe that this section can be shortened and merged with the introduction section.
- The authors provide an algorithm to support clinical decision-making, although this was not indicated in the initial aim of the manuscript. Please revise accordingly.
- The authors do not provide any statement concerning the limitations of their review.
- The authors do not indicate a conclusion/final consideration section. Please revise.
Author Response
English editing:
The manuscript has been reviewed by two native English speakers, and grammatical and typological errors corrected when found.
Reviewer 1
I deeply appreciate the opportunity to review the manuscript titled “To treat or not to treat? That is the question: ventilated patients with multidrug resistant organisms in respiratory specimens”. The authors present a narrative review that aims to “discuss the approach to antimicrobial prescription in ventilated patients who have grown a new MDRO isolates in their respiratory specimens”.
Overall, the manuscript is clear and easy to read. The authors address an interesting topic, especially considering the current COVID-19 pandemic. However, it is unclear to me why is this narrative review required, when the amount of primary evidence supports the conduction of a systematic review. Furthermore, the authors fail to address several other reviews that were published recently on this same topic. Below are my main concerns with the manuscript and suggestions for future revisions.
[Title, abstract and keywords]
Although I do enjoy the spin on Shakespeare’s famous quote, current international guidelines recommend that authors clearly state the study design in the title. This increases potential article finding and citation.
Thank you for this suggestion, changed title to reflect design.
“A narrative review on the approach to antimicrobial use in ventilated patients with multidrug resistant organisms in respiratory samples – to treat or not to treat? That is the question”
The abstract is lacking a more detailed approach to the method, please consider revising it (e.g., used mnemonic, databases/repositories searched, terms, filters, etc.). The abstract does not indicate how many articles were included in this review or the major topics addressed by the included studies. Please revise.
Thank you for this suggestion, we have added a sentence to the abstract and as Reviewer 4 suggested added a Supplementary File to show the search strategy. This sentence was added to the abstract:
“A MEDLINE and PubMed literature search using keywords “multidrug resistant organisms”, “ventilator associated pneumonia” and “decision making”, “treatment” or “strategy “was used to identify 329 references as background for this review.”
Furthermore, a section on the literature search used to inform this review was added:
“A literature search was performed in MEDLINE and PubMed using the strategy de-scribed in Supplementary File 1. The total number of articles identified was 418, of which 89 were duplicates. We reviewed the abstract of 329 articles for inclusion, and utilized the full text of 2 articles considered directly relevant to this narrative review. Following the review of references of identified articles, a further 80 articles were included.”
I would argue that the keywords “critically ill”, “treatment” and “decision” are not required.
Thank you, we have removed “critically ill.” However we believe treatment and decision should remain as keywords. During our literature search, which included “decision” and “treatment”, it was evident that the current literature was lacking detailed discussions on the real-world clinical approach to interpret need to start antimicrobials for ventilated patients with MDROs. Therefore, we believe the narrative review here and the proposed decision framework would be valuable additions
[Introduction]
After reading the introduction section, it is still unclear to me why is this narrative review required – what is the novelty here?
The authors fail to mention several recent reviews that tackle similar topics such as Chaïbi et al. (2022; doi: https://doi.org/10.3390/antibiotics11030359), Carrara et al. (2018; doi: 10.1016/j.ijantimicag.2017.12.013), Abu-Rub et al. (2021; doi: 10.3390/antibiotics10080935), or Pasero et al. (2021; doi: 10.3390/microorganisms9081773). The contributions of this review to existing evidence are lacking.
Thank you for your observation and suggestion. We have reviewed these references and found that they were mostly describing empirical treatment or epidemiology of MDRO in the ICU during the COVID-19 pandemic in general rather than specific to the question of whether treatment is required in ventilated patients with MDRO isolated from the respiratory tract. We have added two of them to the introduction in this sentence to provide more rationale on why we think our review adds value to current literature:
“Recent reviews have provided comprehensive summaries of epidemiological data and the recommended empirical treatment of patients with suspected or confirmed MDRO VAP/VAT [20,29]. However, we believe none explicitly detail, and integrate the different elements of the clinical decision process required to determine whether anti-microbials should be initiated in ventilated patients with known MDRO respiratory isolates.”
However, we did not add the reference by Carrara et al. because it is focused on the nature of empirical treatment, rather than whether to start antibiotics in patients with known MDRO isolates. Furthermore, we did not add the reference by Lubna et al. because it was focused on antibiotics prescription in ICU during the COVID-19 pandemic, rather than being specific to MDRO in the respiratory tract of ventilated patients.
In addition, we have made the aim of the narrative review clearer by stating in the introduction
“Thus the purpose of this narrative review is to discuss the relevant literature supporting the specific set of conditions that must be considered when deciding to initiate antimicrobial therapy in ventilated patients who have grown a new MDRO isolate in their respiratory specimens. As a result of this review, a comprehensive clinical framework is proposed as a guide to facilitate decisions on antimicrobial use in this setting.”
[Method]
There is no indication of a method section, which is rather confusing for a narrative review. A convincing narrative review must be transparent about the sources of information on which the text is based. This is not the case here. Please revise.
Thank you, this has been added right after the introduction under the heading “Literature search”
“A literature search was performed in MEDLINE and PubMed using the strategy de-scribed in Supplementary File 1. The total number of articles identified was 418, of which 70 were duplicates. We reviewed the abstract of 329 articles for inclusion in this narrative review, and utilized the full text of 2 articles considered relevant to this narrative review.”
A supplementary File of the detailed strategy is attached.
The authors fail to provide any statement concerning the database/repositories, search strategy, or filters used. Likewise, it is unclear if study selection, data extraction and report were conducted by a single member or more.
We have now included a literature search for the reader as indicated above. However, because this was a not a systematic review we did not provide detailed analysis on data extraction, or strict study selection criteria. We have included these limitations in the limitation section of the manuscript.
For more information on what is expected in a Narrative Review, please see SANRA - the Scale for the Assessment of Narrative Review Articles, by Baethge et al. (2019, doi: 10.1186/s41073-019-0064-8).
Thank you for your suggestion, we have reviewed SANRA and found that we were particularly deficient in description of literature search and importance of why our review, the addition of which certainly adds value. We have made changes to the introduction and methods section to align with the recommendations from SANRA.
[Results, discussion, and conclusion]
What is the purpose of the second section (Definition of VAP)? I believe that this section can be shortened and merged with the introduction section.
Thank you for this question and suggestion. We believe this section is essential for the reader to consider when deciding whether treatment is required when a MDRO is isolated from a respiratory specimen. However, we share the concern this was not clear to the reviewer and thus have added clarifications for why this section is important.
Establishing whether a patient has VAP/VAT is one of the essential steps in the judgment of the relevance of respiratory MDRO isolates. Patients who are colonized with MDRO do not need treatment and prescription of antimicrobials targeting the MDRO may worsen AMR in this setting [22].
The authors provide an algorithm to support clinical decision-making, although this was not indicated in the initial aim of the manuscript. Please revise accordingly.
Thank you, agree this should have been made clear as we feel the clinical aspects to be considered in the decision making process and this algorithm are the is a major contributions of our manuscript. This has been added to the introduction section:
A comprehensive clinical framework is provided as a guide to facilitate decisions on antimicrobial use in this setting.
The authors do not provide any statement concerning the limitations of their review.
Thank you, this has been added to the new “Limitations” section of the manuscript.
“There are several limitations to narrative reviews that include the subjective nature of the determination of which studies to include, the way the studies are inter-preted, and the conclusions drawn. Nevertheless, our literature search showed that there is limited literature specifically on whether treatment should be given when MDROs are isolated from ventilated patients’ respiratory samples. Although this was not a system-atic review with rigorous data extraction, our aim was to synthesize current best knowledge to provide a clinical framework for facilitate frontline physicians to make best decisions on the need for initiation of antimicrobial use in ventilated patients with MDRO respiratory isolates.”
The authors do not indicate a conclusion/final consideration section. Please revise.
Thank you, this has been added to the conclusion section.
MDRO are commonly isolated from respiratory specimens of mechanically ventilated patients. Treatment decisions for suspected VAP or VAT should be tailored to clinical picture, nature of the specimen, diagnostic confidence, balanced against the cost and risk of worsening AMR. We were unable to identify any clinically focused guidelines directly addressing all aspects of the complex decision-making process required to determine whether the presence of a MDRO specimen requires antimicrobial treatment initiation. We therefore suggest a multi-consideration approach, summarized in a simplified algorithm, to assist clinical decision-making.
Reviewer 2 Report
The topic of antibiotics-1652937 is interesting and fits well the scope of the journals. The reviewer feels it can be accepted after amendments.
1) As a review type manuscript, tables and figures are important. However, this manuscript only has one flow chart type of figure. The authors should consider add more tables and figures, particularly colorful figures.
2) What are the common pathogens cause the MDR in the field? The authors should discuss.
3) The authors should discuss the most recent issue, ie the COVID-19 pandemic. Many patients need to be ventilated.
4) What kind of antibiotics are commonly used to treat these patients?
Author Response
Reviewer 2
The topic of antibiotics-1652937 is interesting and fits well the scope of the journals. The reviewer feels it can be accepted after amendments.
1) As a review type manuscript, tables and figures are important. However, this manuscript only has one flow chart type of figure. The authors should consider add more tables and figures, particularly colorful figures.
Thank you for this suggestion, we added a colored figure on the risk and benefits when balancing whether to start antimicrobial for MDRO in respiratory isolates (Figure 1).
2) What are the common pathogens cause the MDR in the field? The authors should discuss.
Thank you for the suggestion. We agree, however we were only able to add one sentence due to word limit. Furthermore many of the references provided in the manuscript detail the epidemiology of MDROs in VAP/VAT.
Common MDROs in the respiratory tract include multidrug resistant Pseudomonas aeruginosa and Acinetobacter baumannii, which require tigecycline or colistin [13].
3) The authors should discuss the most recent issue, ie the COVID-19 pandemic. Many patients need to be ventilated.
As suggested by reviewer 4 as well, we have added a sentence to the introduction. Unfortunately, word count restrictions do not allow further elaboration and discussion of this issue.
“Furthermore, there are indications that the incidence of VAP caused by MDRO has in-creased during the current COVID-19 pandemic [19].”
4) What kind of antibiotics are commonly used to treat these patients?
Similarly, we are unable to discuss in detail the antibiotics that are given, because of the word limit and since the focus of this review is on whether to give antimicrobials in this review, rather than the antibiotic choice. However, we have added a sentence that addresses this suggestion to some small degree.
“Common MDROs in the respiratory tract include multidrug resistant Pseudomonas aeru-ginosa and Acinetobacter baumannii, which require treatment with antimicrobial agents such as tigecycline or colistin [13].”
Reviewer 3 Report
To treat or not to treat? That is the question: ventilated patients with multidrug resistant organisms in respiratory specimens
The report although is well written, has not fully convinced me on its place in the literature.
Abstract
Multidrug resistant organisms (MDRO) are commonly isolated in respiratory specimens taken from mechanically ventilated patients. The purpose of this narrative review is to discuss the approach to antimicrobial prescription in ventilated patients who have grown a new MDRO isolates in their respiratory specimens. The lack of universally accepted diagnostic criteria for ventilator-associated pneumonia, or ventilator associated tracheobronchitis complicates treatment decisions. Consideration of the clinical context including signs of respiratory infection or deterioration in respiratory or other organ function is essential. The higher the quality of respiratory specimens or the presence of bacteraemia would suggest the MDRO is a true pathogen, rather than colonization, and warrant antimicrobial therapy. A patient with higher severity of illness has lower safety margins and may require initiation of antimicrobial therapy until an alternative diagnosis is established. A structured approach to the decision to treat with antimicrobial therapy is proposed.
Major comment: The abstract denotes that the author intends to propose a treatment guideline of ventilated patients who have grown a new MDRO isolates in their respiratory specimens
However, I reckon such clinical practice guideline is available and reported by organization such as CDC and health authorities:
https://www.cdc.gov/infectioncontrol/guidelines/mdro/index.html
https://www.atsjournals.org/doi/10.1164/rccm.200405-644ST
Minor comment: The title is not clear. in respiratory specimens?
- Definition of VAP
- Characteristics of Respiratory Specimens
- Severity of Illness
- Infection and Inflammatory markers
- Comorbidities
- Alternative diagnoses
- A practical approach to management
Comment: The above topics are rather generic and have been widely reported in literature.
Author Response
Reviewer 3
To treat or not to treat? That is the question: ventilated patients with multidrug resistant organisms in respiratory specimens
The report although is well written, has not fully convinced me on its place in the literature.
Abstract
Multidrug resistant organisms (MDRO) are commonly isolated in respiratory specimens taken from mechanically ventilated patients. The purpose of this narrative review is to discuss the approach to antimicrobial prescription in ventilated patients who have grown a new MDRO isolates in their respiratory specimens. The lack of universally accepted diagnostic criteria for ventilator-associated pneumonia, or ventilator associated tracheobronchitis complicates treatment decisions. Consideration of the clinical context including signs of respiratory infection or deterioration in respiratory or other organ function is essential. The higher the quality of respiratory specimens or the presence of bacteraemia would suggest the MDRO is a true pathogen, rather than colonization, and warrant antimicrobial therapy. A patient with higher severity of illness has lower safety margins and may require initiation of antimicrobial therapy until an alternative diagnosis is established. A structured approach to the decision to treat with antimicrobial therapy is proposed.
Major comment: The abstract denotes that the author intends to propose a treatment guideline of ventilated patients who have grown a new MDRO isolates in their respiratory specimens
However, I reckon such clinical practice guideline is available and reported by organization such as CDC and health authorities:
https://www.cdc.gov/infectioncontrol/guidelines/mdro/index.html
https://www.atsjournals.org/doi/10.1164/rccm.200405-644ST
Thank you for your suggestion, however, we have gone through the references listed and found that for the CDC’s recommendation, it only made one specific suggestion on antimicrobial use in patients with MDRO: “Review the role of antimicrobial use in perpetuating the MDRO problem targeted for intensified intervention. Control and improve antimicrobial use as indicated. Antimicrobial agents that may be targeted include vancomycin, third-generation cephalosporins, and anti-anaerobic agents for VRE, third-generation cephalosporins for ESBLs; and quinolones and carbapenems.” It did not explain what “indicated” means. We believe our review is addressing a separate issue and that is when are antimicrobials “indicated.” In particular, we discuss, and provide an approach to the decision-making process in detail for ventilated patients with MDRO isolates in their respiratory tract.
For the ATS guidelines, the focus is on general VAP rather than MDRO VAP. We believe patients with MDRO present particular challenges in risk and benefit ratio (for patient and other patients) compared to patients with sensitive organisms which complicate the decision making process. This is the purpose of the review.
We have more clearly articulated our aims and importance of this approach in the introduction:
“Recent reviews have provided comprehensive summaries of epidemiological data and the recommended empirical treatment of patients with suspected or confirmed MDRO VAP/VAT [20,29]. However, we believe none explicitly detail, and integrate the different elements of the clinical decision process required to determine whether anti-microbials should be initiated in ventilated patients with known MDRO respiratory isolates. Thus the purpose of this narrative review is to discuss the relevant literature supporting the specific set of conditions that must be considered when deciding to initiate antimicrobial therapy in ventilated patients who have grown a new MDRO isolate in their respiratory specimens. As a result of this review, a comprehensive clinical framework is proposed as a guide to facilitate decisions on antimicrobial use in this setting.”
Minor comment: The title is not clear. in respiratory specimens?
Thank you, we have changed this to sample. The title is now:
“A narrative review on the approach to antimicrobial use in ventilated patients with multidrug resistant organisms in respiratory samples – to treat or not to treat? That is the question”
Definition of VAP
Characteristics of Respiratory Specimens
Severity of Illness
Infection and Inflammatory markers
Comorbidities
Alternative diagnoses
A practical approach to management
Comment: The above topics are rather generic and have been widely reported in literature.
Thank you for this observation, we agree there is much literature about epidemiology of MDRO. However, the detailed thought and specific decision-making process required to decide when patients should be treated is less commonly found addressed in existing literature. Furthermore, discussion on how growth of MDRO complicates this decision process for the patient and the wider ICU setting deserves more attention.
Reviewer 4 Report
The manuscript provides an overview of the approach to antimicrobial prescribing in ventilated patients who have grown a new MDRO isolate in their respiratory samples. The paper is clear and seems to address the topic comprehensively.
However, a description of the literature search is missing. Specifying the search terms and the type of literature included would raise the quality level of the review.
In the introduction, I recommend emphasizing the relationship between the COVID-19 pandemic and increasing risk of healthcare-associated infections. Good references are: Meghan A Baker, Kenneth E Sands, Susan S Huang, Ken Kleinman, Edward J Septimus, Neha Varma, Jackie Blanchard, Russell E Poland, Micaela H Coady, Deborah S Yokoe, Sarah Fraker, Allison Froman, Julia Moody, Laurel Goldin, Amanda Isaacs, Kacie Kleja, Kimberly M Korwek, John Stelling, Adam Clark, Richard Platt, Jonathan B Perlin, CDC Prevention Epicenters Program, The Impact of Coronavirus Disease 2019 (COVID-19) on Healthcare-Associated Infections, Clinical Infectious Diseases, 2021; Baccolini, V., Migliara, G., Isonne, C., Dorelli, B., Barone, L. C., Giannini, D., ... & Villari, P. (2021). The impact of the COVID-19 pandemic on healthcare-associated infections in intensive care unit patients: a retrospective cohort study. Antimicrobial Resistance & Infection Control, 10(1), 1-9; Baker, M. A., Sands, K., Huang, S. S., Kleinman, K., Septimus, E., Varma, N., ... & Perlin, J. B. (2021, November). 171. The Impact of COVID-19 on Healthcare-Associated Infections. In Open Forum Infectious Diseases (Vol. 8, No. Supplement_1, pp. S102-S103). US: Oxford University Press.
Lastly, I recommend a revision of any typographical errors [ For example “posssibly” (line 244), “therpay” (line 248), “localosed” (line 253), “basleine” (line 276)] and a revision of the missing bibliographic references [For example “Antimicrobial resistance (AMR) caused 1.27 million deaths globally in 2019” (line 36)].
Author Response
Reviewer 4
The manuscript provides an overview of the approach to antimicrobial prescribing in ventilated patients who have grown a new MDRO isolate in their respiratory samples. The paper is clear and seems to address the topic comprehensively.
However, a description of the literature search is missing. Specifying the search terms and the type of literature included would raise the quality level of the review.
Thank you for this suggestion, we’ve added a literature search strategy to the abstract, and introduction, and detailed the strategy we used in Supplementary File 1.
In the introduction, I recommend emphasizing the relationship between the COVID-19 pandemic and increasing risk of healthcare-associated infections. Good references are: Meghan A Baker, Kenneth E Sands, Susan S Huang, Ken Kleinman, Edward J Septimus, Neha Varma, Jackie Blanchard, Russell E Poland, Micaela H Coady, Deborah S Yokoe, Sarah Fraker, Allison Froman, Julia Moody, Laurel Goldin, Amanda Isaacs, Kacie Kleja, Kimberly M Korwek, John Stelling, Adam Clark, Richard Platt, Jonathan B Perlin, CDC Prevention Epicenters Program, The Impact of Coronavirus Disease 2019 (COVID-19) on Healthcare-Associated Infections, Clinical Infectious Diseases, 2021; Baccolini, V., Migliara, G., Isonne, C., Dorelli, B., Barone, L. C., Giannini, D., ... & Villari, P. (2021). The impact of the COVID-19 pandemic on healthcare-associated infections in intensive care unit patients: a retrospective cohort study. Antimicrobial Resistance & Infection Control, 10(1), 1-9; Baker, M. A., Sands, K., Huang, S. S., Kleinman, K., Septimus, E., Varma, N., ... & Perlin, J. B. (2021, November). 171. The Impact of COVID-19 on Healthcare-Associated Infections. In Open Forum Infectious Diseases (Vol. 8, No. Supplement_1, pp. S102-S103). US: Oxford University Press.
Thank you for this suggestion, we have added one of the references suggested (the one that makes specific reference to VAP) and made brief mention of the impact of COVID-19 in the introduction:
“Furthermore, during incidence of MDRO VAP has increased in the current COVID-19 pandemic.”
Lastly, I recommend a revision of any typographical errors [ For example “posssibly” (line 244), “therpay” (line 248), “localosed” (line 253), “basleine” (line 276)] and a revision of the missing bibliographic references [For example “Antimicrobial resistance (AMR) caused 1.27 million deaths globally in 2019” (line 36)].
Apologies for the typos, we have gone through the manuscript a few times to remove these typos. Thanks for picking this up.
Round 2
Reviewer 1 Report
Dear authors,
This reviewer is happy with the provided revisions in such a short period of time. Two minor additions could, however, be added in the new method and limitations sections:
- Method: Mention how data extraction and synthesis was performed. Were two independent reviewers involved? If case of a disagreement, was a third reviewer involved in the discussion?
- Limitations: Please briefly refer to the quality of the empirical data so far collected, and potential implications for future research on the topic. Please consider addressing if a systematic review could be conducted given the current status of the literature.
Author Response
Reviewer 1
This reviewer is happy with the provided revisions in such a short period of time. Two minor additions could, however, be added in the new method and limitations sections:
Method: Mention how data extraction and synthesis was performed. Were two independent reviewers involved? If case of a disagreement, was a third reviewer involved in the discussion?
Limitations: Please briefly refer to the quality of the empirical data so far collected, and potential implications for future research on the topic. Please consider addressing if a systematic review could be conducted given the current status of the literature.
Thank you for these suggestions, we agree these are important points for readers to consider when reading our review. We added these to the methods and limitation sections:
Methods:
“A literature search was performed in MEDLINE and PubMed for published
studies since 1910 on March 14, 2022 using the strategy described in
Supplementary File 1. The total number of articles identified was 418, of which
89 were duplicates. All authors independently reviewed the abstract of 329
articles for inclusion, and utilized the full text of 2 articles considered directly
relevant to this narrative review. Following the review of references of identified
articles, 80 additional articles were included. These additional articles were
added after consensus with all authors, in case of disagreement, the
corresponding author made the final decision.”
Limitations:
There are several limitations to narrative reviews that include the subjective
nature of the determination of which studies to include, the way the studies are
interpreted, and the conclusions drawn. Nevertheless, our literature search
showed that there is limited literature specifically on whether treatment should be given when MDROs are isolated from ventilated patients’ respiratory samples.
Specifically, our extensive search did not reveal any high-level evidence to guide
optimal decision making that encompasses and balances all the factors
discussed in this review. While a systematic review of current literature is unlikely
to directly answer the question of antibiotic initiation decision-making, future work should focus on capturing empirical data that can quantify the magnitude of the positive and negative outcomes of antimicrobial treatment in ventilated patients with MDRO in respiratory samples, as summarized in Figures 1 and 2. Although this was not a systematic review with rigorous data extraction by independent reviewers, our aim was to synthesize current best knowledge to provide a clinical framework for facilitate frontline physicians to make best decisions on the need for initiation of antimicrobial use in ventilated patients with MDRO respiratory isolates.
Reviewer 3 Report
“A narrative review on the approach to antimicrobial use in ventilated patients with multidrug resistant organisms in respiratory samples – to treat or not to treat? That is the question”
Suggest to change to : Antimicrobial use in ventilated patients with multidrug resistant organisms isolated from respiratory samples: Clinical Insight
Please add more latest studies related to appropriate use of antibiotic in this targeted group
Author Response
Reviewer 3
A narrative review on the approach to antimicrobial use in ventilated patients with multidrug resistant organisms in respiratory samples – to treat or not to treat? That is the question”
Suggest to change to : Antimicrobial use in ventilated patients with multidrug resistant organisms isolated from respiratory samples: Clinical Insight
Thank you for this suggestion, but we have considered many factors in drafting the latest title, particularly in response to Reviewer 1’s suggestion on following strict PRISMA guidelines. Thus we have modified the title to its current form that reflects the study design, population, clinical setting and research question. We would like to decline the suggestion but would defer this suggestion to the Editor to decide if we should change the title.
Please add more latest studies related to appropriate use of antibiotic in this targeted group
Thank you for this suggestion, we have added new references to show the impact of delayed/inappropriate antibiotics in MDRO VAP. These were added to the manuscript here:
“MDRO generally have worse survival than patients with sensitive pathogens,
especially when inappropriate antibiotics are used [17-19]”
“This is even more important for patients with minimal physiological reserve and
that have high risk of death, as delay in VAP treatment may be fatal [22,84,85].”
Reference 19: Tuon, F.F.; Graf, M.E.; Merlini, A.; Rocha, J.L.; Stallbaum, S.; Arend,
L.N.; Pecoit-Filho, R. Risk factors for mortality in patients with ventilator-associated pneumonia caused by carbapenem-resistant Enterobacteriaceae. Braz J Infect Dis 2017, 21, 1-6, doi:10.1016/j.bjid.2016.09.008.
Reference 85: Puech, B.; Canivet, C.; Teysseyre, L.; Miltgen, G.; Aujoulat, T.; Caron,
M.; Combe, C.; Jabot, J.; Martinet, O.; Allyn, J.; et al. Effect of antibiotic therapy on the prognosis of ventilator-associated pneumonia caused by Stenotrophomonas
maltophilia. Ann Intensive Care 2021, 11, 160, doi:10.1186/s13613-021-00950-1.